# Identifying latent distances with Finslerian geometry

**Alison Pouplin**                                                    ALPU@DTU.DK
*Technical University of Denmark*

**Carl Henrik Ek**                                                    CHE29@CAM.AC.UK
*University of Cambridge*

**David Eklund**                                                    DAVID.EKLUND@RI.SE
*Research Institutes of Sweden*

**Søren Hauberg**                                                    SOHAU@DTU.DK
*Technical University of Denmark*

**Editors:** Sophia Sanborn, Christian Shewmake, Simone Azeglio, Arianna Di Bernardo, Nina Miolane

## Abstract

Riemannian geometry has been shown useful to explore the latent space of generative models. Effectively, we can endow the latent space with the pullback metric obtained from the data space. Because most generative models are stochastic, this metric will be de facto stochastic, and, as a consequence, a deterministic approximation of the metric is required. Here, we are defining a new metric as the expectation of the stochastic curve lengths induced by the pullback metric. We show this metric is, in fact, a Finsler metric. We compare it with a previously studied expected Riemannian metric, and we show that in high dimensions, the metrics converge to each other.

**Keywords:** Finsler geometry, Riemannian geometry, Gaussian Processes, High dimensions.

## 1. Introduction

In machine learning, one objective is to reduce the dimensions of the data using generative models. When this data is assumed to lie near a manifold, we can compute lengths and distances in a low-dimensional latent space using the Riemannian geometry machinery. Such a metric is obtained as the pullback of the immersion $f : \mathcal{Z} \to \mathcal{X}$ that maps the latent space $\mathcal{Z} \subset \mathbb{R}^q$ to the data space $\mathcal{X} \subset \mathbb{R}^D$. In practice, this smooth map $f$ can be the decoding part of a Variational Autoencoder (VAE) (Kingma and Ba, 2014) or a Gaussian Process Latent Variable Model (GP-LVM) (Lawrence, 2003).

One problem is that generative models are stochastic by default, and Riemannian geometry only involves deterministic objects. To dodge this issue, a solution previously studied by Tosi et al. (2014) in the case of GP-LVMs and Arvanitidis et al. (2018) for VAEs was to take the expectation of the Riemannian metric tensor. It has been extensively studied by Hauberg (2018a) as well.

**In this work**, we focus on the quantity that matters to compare data points in the latent space: geodesics. Instead of approximating the metric tensor, we directly approximate the stochastic length by its expectation. This extends the work of Eklund and Hauberg (2019). The findings are the following:

1. We prove that the expected curve length defines a norm and this norm is a Finsler metric.

2. When increasing the number of dimensions, the functionals (curve energy, curve length, volume measure) of the Riemannian and Finsler metrics converge to each other at rate $\mathcal{O}\left(\frac{1}{D}\right)$.

Combined, these findings provide groundings for the previously studied, but heuristically defined, expected Riemannian metric.

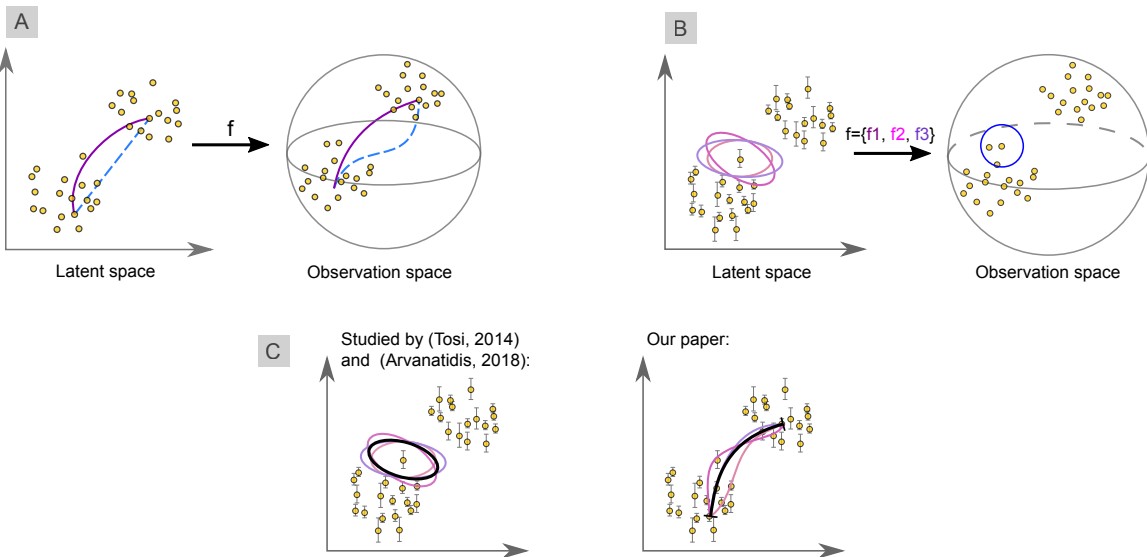

Figure 1: (A) The latent space is mapped to the data space using an immersion $f$. When exploring the latent space with an arbitrary euclidean metric, we don't respect the inherent structure of the data, since the path doesn't follow the manifold. (B) A Riemannian metric is obtained from the immersion $f$. However, $f$ is a stochastic process. Its realisations $f = \{f_1, f_2, f_3\}$ lead to a collection of metrics. (C) The random Riemannian metric is approximated by its expectation. Here, we consider the expectation of the length functional as a Finsler metric.

## 2. Finsler metric as the expected length on a random manifold

The metric pulled-back by a stochastic mapping is stochastic de facto. To compute functionals in the latent space, this metric is approximated by its expectation:

**Definition 1** *(Tosi et al., 2014; Arvanitidis et al., 2018) Let $G$ be a stochastic Riemannian metric tensor. We refer to $\mathbb{E}[G]$ as the **expected metric tensor**, and $g_x : (\mathcal{T}_x\mathcal{M}, \mathcal{T}_x\mathcal{M}) \rightarrow \mathbb{R}_+ : (u, v) \rightarrow u^\top\mathbb{E}[G_x]v$ as the **expected Riemannian metric**, computed at a specific point $x \in \mathcal{M}$. This metric defines a Riemannian metric on $\mathbb{R}^q$.*

Instead of considering the expected Riemannian metric, we aim to define a norm as the expectation of the random paths. The norm of a vector $v$ in $\mathcal{T}_x\mathcal{M}$, induced by a random metric tensor $G$, is $\|v\|_G = \sqrt{v^\top G v}$. We define our new norm as: $v \rightarrow \mathbb{E}\left[\sqrt{v^\top G v}\right]$.

**Proposition 2**  *The function: $F_x : \mathcal{T}_x\mathcal{M} \to \mathbb{R} : v \to \mathbb{E}\left[\sqrt{v^\top G_x v}\right]$ defines a **Finsler metric** but is not induced by a Riemannian metric.*

To compare the expected Riemannian metric with our Finsler metric, we will place ourselves in the specific case where $f$ is a Gaussian process. Gaussian Processes have the advantage of approximately describing the behaviour of deep bayesian neural networks (Matthews et al., 2018; Jacot et al., 2018). The Finsler function can be rewritten in a closed form expression:

**Proposition 3**  *Let $f : \mathbb{R}^q \to \mathbb{R}^D$ be a **Gaussian process**, then:*

$$F_x(v) = \sqrt{2}\sqrt{v^\top \Sigma v}\frac{\Gamma(\frac{D}{2} + \frac{1}{2})}{\Gamma(\frac{D}{2})}{}_1F_1\left(-\frac{1}{2}, \frac{D}{2}, -\frac{\omega}{2}\right),$$

*is the close-form expression of the previously defined Finsler metric at a point $x \in \mathcal{M}$ for a vector $v \in \mathcal{T}_x\mathcal{M}$, with $\omega = (v^\top \Sigma v)^{-1}(v^\top \mathbb{E}[J]^\top \mathbb{E}[J]v)$, $J$ the Jacobian of $f$: $J \sim \prod_{i=1}^{D} \mathcal{N}(\mu_i, \Sigma)$ and ${}_1F_1$ the confluent hypergeometric function of the first kind.*

## 3. Comparison of the Finsler and Riemannian metrics

In geometry, we need to define a metric to compute functionals. In Riemannian geometry, the metric is conveniently obtained by constructing an inner product. The inner product greatly simplifies subsequent computations but it is also restrictive. Finsler geometry is an extension of Riemannian geometry by relaxing the assumption that the metric should be an inner product. Instead the metric is defined as a norm, and so it is not necessarily bilinear or reversible. Ipso facto, Finsler geometry is a generalisation of the Riemannian geometry when we disregard the bilinearity assumption. This difference can be observed by studying indicatrices ($\{v \in \mathcal{T}_x\mathcal{M}, \|v\| = 1\}$), which is a way to see how a metric distorts the space. In the indicatrix-field represented in Figure 2: we see that the Finsler indicatrices can have almost a rectangular shape while the Riemannian ones are ellipses.

In the area of low variance (near the data points), the Finsler and Riemannian indicatrices are alike. This is explained because the metrics differ by a variance term. If our mapping $f$ was deterministic, both metrics would agree. In general, we will see that in high dimensions, they converge to each other:

**Proposition 4**  *The relative difference between the Finsler metric $F_x : v \to \mathbb{E}[\sqrt{v^\top G_x v}]$ and the Riemannian metric $g_x : (v, v) \to v^\top \mathbb{E}[G_x]v$ is:*

$$0 \leq \frac{\sqrt{g_x(v,v)} - F_x(v)}{\sqrt{g_x(v,v)}} \leq \frac{\mathrm{Var}[v^\top G_x v]}{2\mathbb{E}[v^\top G_x v]^2}.$$

*This proposition is valid for any stochastic immersion.*

In this proposition, we are comparing the norms induced by their respective metrics. The Finsler metric is a norm by definition, while the Riemannian metric is defined as an inner product, which defines a norm by taking its squareroot. These metrics become equal in two cases: when the variance converges to zero, and when the number of dimensions increases. The latter case is investigated below for a Gaussian process.

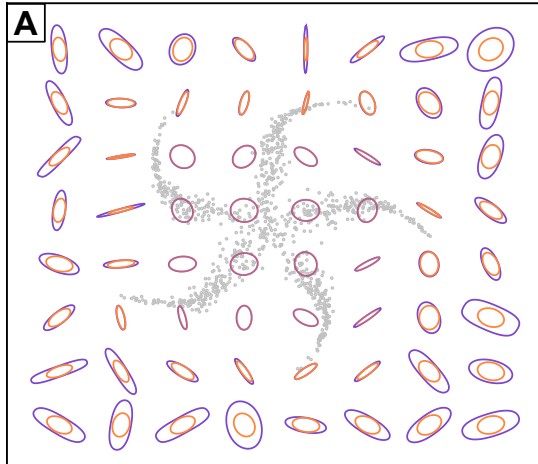 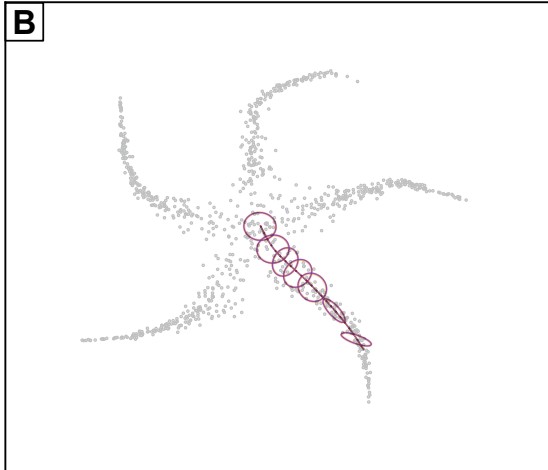

Figure 2: Indicatrix-field over the latent space of of pinwheel data (grey) representing the Riemannian (orange) and Finslerian (purple) metrics. The indicatrices are computed: (A) over a grid in the latent space, (B) along a geodesic.

**Proposition 5** *When $f : \mathbb{R}^q \to \mathbb{R}^D$ is a Gaussian process, the relative ratio becomes:*

$$0 \leq \frac{\sqrt{g_x(v,v)} - F_x(v)}{\sqrt{g_x(v,v)}} \leq \frac{1}{D+\omega} + \frac{\omega}{(D+\omega)^2},$$

*with $\omega = (v^\top \Sigma v)^{-1}(v^\top \mathbb{E}[J]^\top \mathbb{E}[J]v)$ and $J$ the Jacobian of $f$: $J \sim \prod_{i=1}^{D} \mathcal{N}(\mu_i, \Sigma)$.*

Under the assumption that $\omega$ is lower bounded and we are working on a compact set, we can find upper bounds for all the functionals derived from the Finsler and Riemannian metrics, in particular: the curve lengths ($\mathcal{L}$ and $L_G$), the curve energy($\mathcal{E}$ and $E_G$) and the volume measure ($\mathcal{V}$ and $V_G$):

**Proposition 6** *In high number of dimensions, we have:*

$$\frac{L_G(x) - \mathcal{L}(x)}{L_G(x)} = \mathcal{O}\left(\frac{1}{D}\right), \qquad \frac{E_G(x) - \mathcal{E}(x)}{E_G(x)} = \mathcal{O}\left(\frac{1}{D}\right), \qquad \frac{V_G(x) - \mathcal{V}(x)}{V_G(x)} = \mathcal{O}\left(\frac{q}{D}\right),$$

*and, when $D$ converges toward infinity: $L_G \sim \mathcal{L}, E_G \sim \mathcal{E}$ and $V_G \sim \mathcal{V}$.*

## 4. Discussion

Generative models are often used to reduce data dimension in order to better understand the mechanisms behind the data generating process. We consider the general setting where the mapping from latent variables to observations is driven by a smooth stochastic process, and the sample mappings span Riemannian manifolds. The Riemannian geometry machinery has already been used in the past to explore the latent space.

The Finslerian view of the latent representation gives us a suitable general solution to explore a random manifold, but it does not immediately translate into a practical computational tool. As Riemannian manifolds are better understood computationally than Finsler

manifolds, we have raised the question: How good an approximation of the Finsler metric can be achieved by a Riemannian metric? The answer turns out to be: quite good. We have shown that as data dimension increases, the Finsler metric becomes increasingly Riemannian. We have a justification for approximating the Finsler metric with a Riemannian metric such that computational tools become more easily available. In practice we find that geodesics under the Finsler and the Riemannian metric are near identical except in regions of high uncertainty.

**Acknowledgments.** This work was funded in part by the Novo Nordisk Foundation through the Center for Basic Machine Learning Research in Life Science (NNF20OC0062606). It also received funding from the European Research Council (ERC) under the European Union's Horizon 2020 research, innovation programme (757360). SH was supported in part by research grants (15334, 42062) from VILLUM FONDEN. The authors also acknowledge the support of the Pioneer Centre for AI, DNRF grant number P1.

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

## Appendix A. Notations and definitions

We define $f : \mathbb{R}^q \to \mathbb{R}^D$ a stochastic immersion (when specified, a Gaussian process), $J$ its Jacobian, and $G_x = J^\top J$ the stochastic Riemannian metric tensor at a point $x$ of the manifold. Key functionals are also investigated: the curve length, the curve energy and the volume measure, denoted $L_G$, $E_G$ and $V_G$ for the expected Riemannian metric and $\mathcal{L}$, $\mathcal{E}$ and $\mathcal{V}$ for the Finsler metric. The volume measure allows for integrating probabilities over the manifold, while the energy functionals (equivalently, the length functionals) are needed to compute geodesics.

| | |
|---|---|
| $F$ | Finsler metric: $F : (\mathcal{M}, \mathcal{T}_x\mathcal{M}) \to \mathbb{R}$ |
| $\mathcal{M}$ | Smooth differentiable manifold |
| $\mathcal{T}_x\mathcal{M}$ | Tangent space of the manifold $\mathcal{M}$ at a point $x$ |
| $D$ | Number of dimensions of the observational space |
| $q$ | Number dimensions of the latent space |
| $J_f$ | Jacobian of a stochastic function $f : \mathbb{R}^q \to \mathbb{R}^D$ |
| $\Sigma_x$ | Covariance matrix of the Jacobian $J_x \sim \prod_{i=1}^{D} \mathcal{N}(\mu_i, \Sigma_x)$ |
| $G_x$ | Stochastic metric tensor defined as; $G_x = J_x^\top J_x$ |
| $\mathcal{L}, \mathcal{E}, \mathcal{V}$ | Length, energy and Busemann Hausdorff volume of the Finsler metric |
| $L_G, E_G, V_G$ | Length, energy and volume of the Riemannian metric with $\mathbb{E}[G]$ the expected metric tensor. |

**immersion**    An immersion $f : \mathcal{M} \to \mathbb{R}^D$ is a smooth map such that its derivative $\partial f : \mathcal{T}_x\mathcal{M} \to \mathcal{T}_{f(x)}\mathbb{R}^D$ is injective at each point $x \in \mathcal{M}$.

**indicatrix**    An indicatrix is a befitted way to represent the distortion induced by the metric on a unit circle. Rigorously speaking, for a given point on the manifold, an indicatrix of a metric is defined as the set of vectors in the tangent space such that norm obtained from the metric is equal to one: $\{v \in \mathcal{T}_x\mathcal{M}, ||v|| = 1\}$. If our metric is euclidean, we will only have a linear transformation between the latent and the observational spaces, and the indicatrix would still be a circle. Because the Riemannian metric is quadratic, it will always generate an ellipse in the latent space. The Finsler indicatrix, however, would have a convex, even asymetrical, shape.

**curve length** We consider a curve $\gamma(t)$ and its derivative $\dot{\gamma}(t)$ on a manifold $\mathcal{M}$ equipped with a norm $\|\cdot\|_m$ endowed by a metric $m$ – a Finsler metric or a Riemannian metric. We define the length of the curve: $L(\gamma) = \int \|\dot{\gamma}(t)\|_m \, dt$.

**geodesic** Locally length-minimising curves between two connecting points.

**energy curve** We define the energy of the curve: $E(\gamma) = \int \|\dot{\gamma}(t)\|_m^2 \, dt$.

**volume measure** The volume measure, or volume form, is a quantity that enables us to integrate over a manifold and accounts for the amount of distortion induced by the metric. In Riemannian geometry, the volume measure is uniquely defined: $V_G = \sqrt{\det G}$. In Finsler geometry, the volume has different definition. In this work, we are using the Busemann-Hausdorff definition: $\mathcal{V} = \mathrm{vol}(\mathbb{B}^n(1))/\mathrm{vol}(\{v \in \mathcal{T}_x\mathcal{M}|F_x(v) < 1\})$.

## Appendix B. Proofs

**Lemma 7** *The Hessian matrix $\frac{1}{2}Hess(F(x,v)^2)$ of the function $F(x,v) = \mathbb{E}\left[\sqrt{v^\top G_x v}\right]$ is equal to:*

$$\frac{1}{2}Hess(F(x,v)^2) = \mathbb{E}\left[(v^\top G_x v)^{\frac{1}{2}}\right] \mathbb{E}\left[(v^\top G_x v)^{-\frac{1}{2}}G_x - (v^\top G_x v)^{-\frac{3}{2}}G_x vv^\top G_x\right] + \mathbb{E}\left[(v^\top G_x v)^{-\frac{1}{2}}G_x\right]^2 vv^\top$$

**Proof** Let $g : \mathbb{R}^q \to \mathbb{R} : v \to \sqrt{v^\top G v}$ be a scalar-value stochastic function that takes as input a column vector $v$, and $G$ a random symmetric matrix. We would like to know the different derivatives of $g$ with respect to $v$. We name by default $J_g$ and $H_g$, its Jacobian and Hessian matrix. Using the chain rule, we have: $J_g = (v^\top G v)^{-\frac{1}{2}}Gv$ and $H_g = (v^\top G v)^{-\frac{1}{2}}G - (v^\top G v)^{-\frac{3}{2}}(Gvv^\top G)$.

For the rest of the proof, we need to show that we can invert the derivatives with the expectation: $\frac{\partial \mathbb{E}[g]}{\partial v_i} = \mathbb{E}[\frac{\partial g}{\partial v_i}]$. The sequence of functions $(\frac{\partial g}{\partial v_i})$ being always upper bounded by $Gv_i$ and $\mathbb{E}[Gv_i] < \infty$, using the theorem of dominated convergence, we can invert the expectation with the derivatives. Finding the Jacobian of the expectation of the function $g$ is the same than finding the expectation of the Jacobian $J_g$, and similarly for the Hessian $H_g$.

We can then write, for $F : \mathbb{R}^q \to \mathbb{R} : \mathbb{E}[v \to \sqrt{v^\top G v}]$, then

$$\text{Hess}(F) = \mathbb{E}[H_g] = \mathbb{E}\left[(v^\top G v)^{-\frac{1}{2}}G - (v^\top G v)^{-\frac{3}{2}}Gvv^\top G\right]$$

$$\nabla F = \mathbb{E}[J_g] = \mathbb{E}[(v^\top G v)^{-\frac{1}{2}}Gv]$$

.

We now consider the function $h : \mathbb{R}^q \to \mathbb{R} : v \to \mathbb{E}[\sqrt{v^\top G v}]^2 = F(v)^2$. Using the chain rule again, and inverting expectation and derivatives, we have its Hessian

$$H_h = F \cdot \text{Hess}[F] + \nabla F^\top \nabla F = 2\mathbb{E}[g(v)]\mathbb{E}[H_g] + 2\mathbb{E}[J_g]^\top \mathbb{E}[J_g].$$

Finally, replacing $J_g$ and $H_g$ previously obtained in this expression, we conclude:

$$\frac{1}{2}H_h = \mathbb{E}\left[(v^\top Gv)^{\frac{1}{2}}\right]\mathbb{E}\left[(v^\top Gv)^{-\frac{1}{2}}G - (v^\top Gv)^{-\frac{3}{2}}Gvv^\top G\right] + \mathbb{E}\left[(v^\top Gv)^{-\frac{1}{2}}G\right]^2 vv^\top.$$

■

**Proposition 8**  *The function:* $F_x : \mathcal{T}_x\mathcal{M} \rightarrow \mathbb{R} : v \rightarrow \mathbb{E}\left[\sqrt{v^\top G_x v}\right]$ *defines a **Finsler metric** but is not induced by a Riemannian metric.*

**Proof**  First, we need to prove the function $F_x(v) = \mathbb{E}\left[\sqrt{v^\top G_x v}\right]$ is:

1. **positive homogeneous**: $\forall \lambda \in \mathbb{R}_+, F(p, \lambda v) = \lambda F_x(v)$

2. **smooth**: $F_x(v)$ is a $C^\infty$ function on the slit tangent bundle $\mathcal{TM} \setminus \{0\}$

1) Let $\lambda \in \mathbb{R}$, then we have: $F(x, \lambda v) = \mathbb{E}\left[\sqrt{\lambda^2 v^\top G_x v}\right] = |\lambda|\left[\sqrt{v^\top G_x v}\right]$.

2) The multivariate function: $\mathbb{R}^q \setminus \{0\} \rightarrow \mathbb{R}_+^* : v \rightarrow v^\top G_x v$ is $C^\infty$ and strictly positive, since $G_x = J_f^\top J_f$ is positive definite. The function $\mathbb{R}_+^* \rightarrow \mathbb{R}_+^* : x \rightarrow \sqrt{x}$ is also $C^\infty$. Finally, $\mathbb{R}_+^* \rightarrow \mathbb{R}_+^* : x \rightarrow \mathbb{E}[x]$ is by definition differentiable. By composition, $F_x(v)$ is a $C^\infty$ function on the slit tangent bundle $\mathcal{TM} \setminus \{0\}$.

Second, we need to prove the function $F_x(v) = \mathbb{E}\left[\sqrt{v^\top G_x v}\right]$ satisfies **strong convexity criterion**:

Proving that F satisfies the strong convexity criterion is equivalent to show that the Hessian matrix $H = \frac{1}{2}\text{Hess}(F_x(v)^2)$ is strictly positive definite. Thus, we need to prove that $\forall w \in \mathbb{R}^q \setminus \{0\}, w^\top H w > 0$. According to Lemma 7, because the expectation is a positive function, it's straightforward to see that $\forall w \in \mathbb{R}^q \setminus \{0\}, w^\top H w \geq 0$. The tricky part of this proof is to show that $w^\top H w > 0$. This can be obtained if one of the terms ($F \cdot \text{Hess}(F)$ or $\nabla F^\top \nabla F$) is strictly positive.

First, let's decompose $H$ as the sum of matrices: $H = F\text{Hess}(F) + \nabla F^\top \nabla F$ (Lemma 7), with:

$$F \cdot \text{Hess}(F) = \mathbb{E}\left[(v^\top Gv)^{\frac{1}{2}}\right]\mathbb{E}\left[(v^\top Gv)^{-\frac{3}{2}}\left((v^\top Gv)G - Gv(Gv)^\top\right)\right],$$

$$\nabla F^\top \nabla F = \mathbb{E}\left[(v^\top Gv)^{-\frac{1}{2}}G\right]^2 vv^\top.$$

We will study two cases: when $w \in \text{span}(v)$, and when $w \notin \text{span}(v)$. We will always assume that $v \neq 0$, and so by definition: $F_x(v) > 0$.

Let $w \in \text{span}(v)$. We will show that $w^\top \nabla F^\top \nabla F w > 0$. We have $w = \alpha v, \alpha \in \mathbb{R}$. Because F is 1-homogeneous and using Euler theorem, we have: $\nabla F_x(v)v = F_x(v)$. Then $(\alpha v)^\top \nabla F^\top \nabla F(\alpha v) = \alpha^2 F^2$, and $\alpha^2 F_x(v)^2 > 0$.

Let $w \notin \text{span}(v)$. F being a scalar function, we have: $w^\top F\text{Hess}[F]w = Fw^\top \text{Hess}[F]w$. We would like to show that: $w^\top \text{Hess}[F]w > 0$. The strategy is the following: if we prove that the kernel of $\text{Hess}[F]$ is equal to the span$(v)$, then $w \notin \text{span}(v)$ is equivalent to say that $w \notin \text{ker}(\text{Hess}[F])$ and we can conclude that: $w^\top \text{Hess}[F]w > 0$. Let's prove span$(v) \in$

ker(Hess($F$)). We know that Hess($F$)$v = 0$, since F is 1-homogeneous, so we have span($v$) $\in$ ker(Hess($F$)). To obtain the equality, we just need to prove that the dimension of the kernel is equal to 1. Let $z \in \text{span}(v^\top G)^\top$, which is $(Gv)^T z = 0$. We have $\dim(\text{span}(v^\top M)) = 1$, and thus: $\dim(\text{span}(v^\top G)^\top) = q - 1$. Furthermore, $z^\top \text{Hess}[F] z = z^\top \mathbb{E}\left[M(v^\top Mv)^{-\frac{1}{2}}\right] z > 0$, so we can deduce that $\dim(\text{im}(\text{Hess}[F])) = q - 1$. Using the Rank-Nullity theorem, we conclude that $\dim(\ker(\text{Hess}(F))) = q - \dim(\text{im}(\text{Hess}[F])) = 1$, which concludes the proof.

In conclusion, $\forall w \in \mathbb{R}^q \backslash \{0\}, w^\top \frac{1}{2}\text{Hess}(F_x(v)^2)w > 0$. The function $F$ satisfies the strong convexity criterion. ∎

**Proposition 9** *Let $f$ be a **Gaussian process**, then:*

$$F_x(v) = \sqrt{2}\sqrt{v^\top \Sigma_x v}\frac{\Gamma(\frac{D}{2} + \frac{1}{2})}{\Gamma(\frac{D}{2})}{}_1F_1\left(-\frac{1}{2}, \frac{D}{2}, -\frac{\omega}{2}\right),$$

*with $\omega = (v^\top \Sigma_x v)^{-1}(v^\top \mathbb{E}[J]^\top \mathbb{E}[J]v)$ and ${}_1F_1$ the confluent hypergeometric function of the first kind.*

**Proof** The objective of the proof is to show that, if the Jacobian $J_f$ follows a non-central normal distribution, then, $\forall v \in \mathbb{R}^q$, the expectation $\mathbb{E}[v^\top J_f^\top J_f v]$ will follow a non-central Nakagami distribution. This is a particular case of the derivation of moments of non-central Wishart distributions, previously shown and studied by Kent and Muirhead (1984); Hauberg (2018b).

By hypothesis, $J_f$ follows a non-central normal distribution: $J_f \sim \mathcal{N}(\mathbb{E}[J], I_D \otimes \Sigma_x)$. Then, $G_x = J_f^\top J_f$ follows a non-central Wishart distribution: $G_x \sim \mathcal{W}_d(D, \Sigma_x, \Sigma_x^{-1}\mathbb{E}[J]^\top \mathbb{E}[J])$. According to (Kent and Muirhead, 1984, Theorem 10.3.5.), $v^\top G_x v$ will also follow a non-central Wishart distribution: $v^\top G_x v \sim \mathcal{W}_1(D, v^\top \Sigma_x v, \omega)$, with: $\omega = (v^\top \Sigma_x v)^{-1}(v^\top \mathbb{E}[J]^\top \mathbb{E}[J]v)$.

To compute $\mathbb{E}[\sqrt{v^\top G_x v}]$, we shall look at the derivation of moments. (Kent and Muirhead, 1984, Theorem 10.3.7.) states that: if $X \sim \mathcal{W}_q(D, \Sigma, \Omega')$, with $q \leq D$, then $\mathbb{E}[(\det(X))^k] = (\det \Sigma)^k 2^{qk}\frac{\Gamma_q(\frac{D}{2}+k)}{\Gamma_q(\frac{D}{2})}{}_1F_1(-k, \frac{D}{2}, -\frac{1}{2}\Omega')$. We directly apply the theorem to our case, knowing that $v^\top G_x v$ is a scalar term, so $\det(v^\top G_x v) = v^\top G_x v$, $q = 1$, and $k = \frac{1}{2}$:

$$\mathbb{E}[\sqrt{v^\top G_x v}] = \sqrt{2}\sqrt{v^\top \Sigma v}\frac{\Gamma(\frac{D}{2} + \frac{1}{2})}{\Gamma(\frac{D}{2})}{}_1F_1(-\frac{1}{2}, \frac{D}{2}, -\frac{1}{2}\omega)$$

∎

**Proposition 10** *The relative difference between the Finsler metric: $F_x : v \to \mathbb{E}[\sqrt{v^\top G_x v}]$ and the Riemmanian metric $g : (v, v) \to v^\top \mathbb{E}[G_x]v$ is:*

$$0 \leq \frac{\sqrt{g(v, v)} - F_x(v)}{\sqrt{g(v, v)}} \leq \frac{\text{Var}[v^\top G_x v]}{2\mathbb{E}[v^\top G_x v]^2}.$$

*This proposition is valid for any stochastic immersion.*

**Proof** We will directly use a sharpen version of Jensen's inequality obtained by Liao and Berg (2019): Let X be a one-dimensional random variable with mean $\mu$ and $P(X \in (a,b)) = 1$, where $-\infty \leq a \leq b \leq +\infty$. Let $\phi$ a twice derivable function on $(a,b)$. We further define: $h(x,\mu) = \frac{\phi(x)-\phi(\mu)}{(x-\mu)^2} - \frac{\phi'(\mu)}{x-\mu}$. Then:

$$\inf_{x \in (a,b)} \{h(x,\mu)\}\mathrm{Var}[X] \leq \mathbb{E}[\phi(x)] - \phi(\mathbb{E}[x]) \leq \sup_{x \in (a,b)} \{h(x,\mu)\}\mathrm{Var}[X].$$

In our case, we will chose $\phi : z \to \sqrt{z}$ with $z$ a one-dimensional random variable defined as $z = v^\top G_x v$. $a = 0$, $b = +\infty$ and $\mu = \mathbb{E}[z]$. $h(z,\mu) = (\sqrt{z}-\sqrt{\mu})(z-\mu)^{-2} - (2(z-\mu)\sqrt{\mu})^{-1}$. Because its first derivative $\phi'$ is convex, the function $x \to h(x,\mu)$ is monotonically increasing. Thus:

$$\inf_{z \in (0,+\infty)} \{h(x,\mu)\} = \lim_{z \to 0} = -\frac{\sqrt{\mu}}{2\mu^2} \quad \text{and} \quad \sup_{z \in (0,+\infty)} \{h(x,\mu)\} = \lim_{z \to +\infty} = 0.$$

It finally gives:

$$-\frac{\sqrt{\mu}}{2\mu^2}\mathrm{Var}[z] \leq \mathbb{E}[\sqrt{z}] - \sqrt{\mathbb{E}[z]} \leq 0.$$

Replacing $F(x,v) = \mathbb{E}[\sqrt{z}]$ and $\sqrt{g(v)} = \sqrt{\mathbb{E}[z]} = \sqrt{\mu}$ concludes the proof. ∎

**Proposition 11** *When $f$ is a Gaussian process, the relative ratio becomes:*

$$0 \leq \frac{\sqrt{g(v,v)} - F_x(v)}{\sqrt{g(v,v)}} \leq \frac{1}{D+\omega} + \frac{\omega}{(D+\omega)^2}.$$

**Proof** (Kent and Muirhead, 1984, Theorem 10.3.7.) states that if $z \sim \mathcal{W}_1(D,\sigma,\omega)$ then $\mathbb{E}[z^k] = \sigma^k 2^k \frac{\Gamma(\frac{D}{2}+k)}{\Gamma(\frac{D}{2})} {}_1F_1(-k,\frac{D}{2},-\frac{1}{2}\Omega)$. In particular, for $k=1$ and $k=2$, we have ${}_1F_1(-1,b,c) = 1 - \frac{c}{b}$ and ${}_1F_1(-2,b,c) = 1 - \frac{2c}{b} + \frac{c^2}{b(b+1)}$. We also have $\frac{\Gamma(\frac{D}{2}+1)}{\Gamma(\frac{D}{2})} = \frac{D}{2}$ and $\frac{\Gamma(\frac{D}{2}+2)}{\Gamma(\frac{D}{2})} = \frac{D}{2}\left(\frac{D}{2}+1\right)$, which leads to: $\mathbb{E}[z] = \sigma(D+\Omega)$ and $\mathbb{E}[z^2] = \sigma^2(2\omega + 2(D+\omega) + (D+\omega)^2)$. Finally, we conclude:

$$\frac{\mathrm{Var}[z]}{\mathbb{E}[z]^2} = \frac{\mathbb{E}[z^2]}{\mathbb{E}[z]^2} - 1 = \frac{2\omega}{(D+\omega)^2} + \frac{2}{D+\omega}.$$

Knowing that $z = v^\top G_x v$, and with the proposition: $0 \leq \frac{\sqrt{g(v,v)} - F_x(v)}{\sqrt{g(v,v)}} \leq \frac{\mathrm{Var}[v^\top G_x v]}{2\mathbb{E}[v^\top G_x v]^2}$, we have the desired result. ∎

**Lemma 12** *Our finsler metric $v \to \mathbb{E}[\sqrt{v^\top G_x v}]$ is defined with $v^\top G_x v \sim \mathcal{W}_1(D, v^\top \Sigma_x v, \omega)$, and $\omega = (v^\top \Sigma_x v)^{-1}(v^\top \mathbb{E}[J]^\top \mathbb{E}[J]v)$. We assume that $\mathbb{E}[J]$ is upper bounded. Let $M \in \mathbb{R}_{+,}$. We have:*

$$\omega \leq DM$$

**Proof**  By definition, $\Sigma$ does not depend on $D$. We assume that every element of the expected jacobian is upper bounded: $\mathbb{E}[J]_{ij} \leq m$, with $m \in \mathbb{R}_+^*$. We call $\sigma = v^\top \Sigma_x v \in \mathbb{R}_+^*$. We have:

$$\omega = \sigma^{-1} \sum_{k=1}^{D} \sum_{i=1}^{q} \sum_{j=1}^{q} v_i \mathbb{E}[J]_{ki} \mathbb{E}[J]_{kj} v_j \leq \sigma^{-1} \sum_{k=1}^{D} m^2 \|v\|^2 \leq DM,$$

with $M = \sigma^{-1} m^2 \|v\|^2 \in \mathbb{R}_+^*$, and M does not depend on D. ∎

**Proposition 13**  *In high number of dimensions, we have:*

$$\frac{L_G(x) - \mathcal{L}(x)}{L_G(x)} = \mathcal{O}\left(\frac{1}{D}\right), \qquad \frac{E_G(x) - \mathcal{E}(x)}{E_G(x)} = \mathcal{O}\left(\frac{1}{D}\right), \qquad \frac{V_G(x) - \mathcal{V}(x)}{V_G(x)} = \mathcal{O}\left(\frac{q}{D}\right).$$

*And, when D converges toward infinity: $L_G \sim \mathcal{L}, E_G \sim \mathcal{E}$ and $V_G \sim \mathcal{V}$.*

**Proof**  We assume that $\omega$ is lower bounded by $m \in \mathbb{R}^+$. Using lemma 12, it is also upper bounded by $M \in \mathbb{R}+$. We can show that:

Thus:

$$\frac{L_G(x) - \mathcal{L}(x)}{L_G(x)} \leq \frac{m_1}{D}$$

$$\frac{E_G(x) - \mathcal{E}(x)}{E_G(x)} \leq \frac{m_2}{D}$$

$$\frac{V_G(x) - \mathcal{V}(x)}{V_G(x)} \leq 1 - \left(1 - \frac{m_3}{D}\right)^q$$

with $m_1, m_2, m_3 \in \mathbb{R}_+$.

Using Taylor series expansion, when $x \sim 0$, we have: $1 - (1 - x)^q = qx + o(x^2)$. Thus:

$$\frac{V_G(x) - \mathcal{V}(x)}{V_G(x)} \leq m_3 \frac{q}{D}.$$

We also assume that our space is compact. Using Heine Borel theorem, the metrics and their functional are bounded. In high dimensions, all functionals converge to each others. This concludes the proof. ∎

## Appendix C. Some experiments

We want to illustrate cases where those metrics could be useful in practice, for real world data. For this we use three datasets (a synthetic dataset, composed of data representing a pinwheel projected onto a sphere), a font dataset Campbell and Kautz (2014). We trained a GP-LVM model to learn a 2d-manifold. From the optimised Gaussian process, we can access the Riemannian and Finsler metric, and minimise their respective curve energies to obtain geodesics.

As we can see, the Finsler and Riemannian geodesics coincide in all cases. For all latent spaces (A.1. and B.1. in Figure 3, The heatmap represents the Riemannian volume measure in logarithm scale. The volume measure is low in area of high density and high in area of low density of data points.

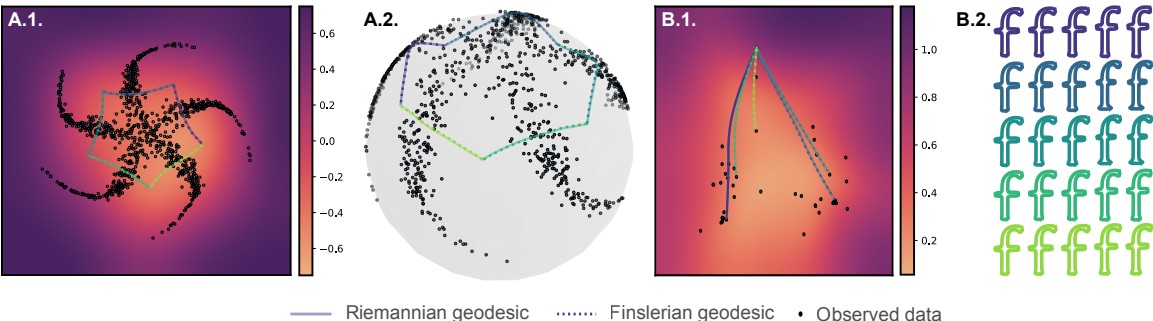

Figure 3: Geodesics computed for latent (A.1., B.1.) and observational (A.2., B.2.) spaces. (A) The dataset used was a pinwheel projected onto a sphere, as seen in A.2. (B) The dataset consists of the position of the markers parametrising the contour of the letter **f**.

