# OpenReview forum: "Identifying latent distances with Finslerian geometry"
_NeurIPS.cc/2022/Workshop/NeurReps — NeurReps 2022 Poster_

### Official Review · Reviewer_pFR3 · 2022-10-05
**Propose and study expected Finsler metric instead of expected Riemannian metric for geometric modeling of latent spaces of generative models**

**Confidence:** 4
**Soundness:** 4
**Presentation:** 3
**Contribution:** 3
**Overall Rating:** 6

**Summary:**

Define an expected (symmetric) Finsler metric from stochastic immersions (like Gaussian processes) instead of expected Riemannian metric for geometric modeling of latent spaces of generative models. Show that in low variance regions, the Finsler and Riemannian metrics converge but differ significantly in high variance regions.

**Questions:**

- In general, Finsler distances may be asymmetric but here we have symmetric norms and symmetric distances. Mention this.
- Other stochastic immersions different than GPs?
-Question: Is there convergence results a la Gromov-Haussdorf of the expected Finsler spaces?
-Would be better to add another geodesic with different endpoints to have something different and explain the low variance near data points.


**Limitations:**

More details and experiments (like Appendix C) would be valuable.

**Recommended Decision:**

3: Accept

**Relevance:**

3: Solid fit

**Strengths And Weaknesses:**

Strength:
- original research topic, interesting topic for the workshop.
- Solid contributions are in Propositions 2-6 and details in the appendix.

Weaknesses::
- More background on the expected Riemannian metric would improve the reading of the paper.
- The difference between Finsler and Riemannian indicatrices on the curve of Fig 2B is not much: add also other curves with indicatrices
- Maybe cite more refs in medical imaging apps using Finsler geometry

Typos:
- page 1  converge to each other at RATE O()
- Def 1. computed AT a specific
- page 3 both metricS would agree
- check bibliography jensen->Jensen, incomplete refs for Eklund et al. an Tosi et al.

**Submission Track:**

Extended Abstract (4 Page)

---

> ### Author Response · Authors · 2022-10-27
> **Response to Reviewer pFR3**
>
> We would like to thank the reviewer for the very constructive feedback, and answer some of their questions below:
>
> * `Maybe cite more refs in medical imaging apps using Finsler geometry`.
>
> Do you have some references in medical imaging applications? We could add those in camera ready version to motivate the use of Finsler geometry.
>
> * `In general, Finsler distances may be asymmetric but here we have symmetric norms and symmetric distances. Mention this.`
>
> Absolutely, this will be added in the camera-ready version.
>
> * `Other stochastic immersions different than GPs?`
>
> Most of the results obtained in this abstract assume that the immersion is, in particular, a Gaussian Process, except for Proposition 4 which is valid for any stochastic process. We can argue that, in general, the decoder of a VAE is a stochastic immersion. On a side note, this general case has been studied by [1].
>
> * ` Is there convergence results a la Gromov-Haussdorf of the expected Finsler spaces? `
>
> This is an excellent question, thank you for pointing this out. While the idea of investigating other convergences is interesting, we believe that the Gromov-Haussdorff convergence in particular might be a weaker result that what is shown in this extended abstract. For example, two Riemannian manifolds might converge in the GH sense but their respective functionals, such as the volume measure, can differ [2].
>
> * `Would be better to add another geodesic with different endpoints to have something different `
>
> We might not be able to do more experiments for the camera-ready version, but we will definitely keep it in mind for the poster session.
>
> * `and explain the low variance near data points.`
>
> Indeed, in the paper, we stated that "*In the area of low variance (near the data points), the Finsler and Riemannian indicatrices are alike. This is explained because the metrics differ by a variance term*". This can retreive from the work of [3] that: $E_G (\gamma) = \mathcal{E} (\gamma) + \int_0^1 \text{var} \left[ ||\dot{\gamma}(t)^{\top} G \dot{\gamma}(t)|| \right]dt $.
>
> Also, in general, $ \text{var} \left[ || \dot{\gamma}(t)^{\top} G \dot{\gamma}(t) || \right] \leq \text{var} \left[ \dot{\gamma}(t)^{\top} G \dot{\gamma}(t) \right]$, and the posterior variance of a Gaussian process, when the kernel is stationary, tend to be low when the point is within the support of the data (at least, compared to a point outside of the support of the data). This will be clarified in the extended abstract.
>
> ---
> [1] Arvanitidis, Georgios, et al. "Pulling back information geometry." arXiv preprint arXiv:2106.05367 (2021).
>
> [2] Sormani, Christina. "How Riemannian manifolds converge." Metric and differential geometry. Birkhäuser, Basel, 2012. 91-117.
>
> [3] Eklund, David, and Søren Hauberg. "Expected path length on random manifolds." arXiv preprint arXiv:1908.07377 (2019).

---

### Official Review · Reviewer_ssry · 2022-10-13
**Not self-contained**

**Confidence:** 3
**Soundness:** 3
**Presentation:** 2
**Contribution:** 2
**Overall Rating:** 3

**Summary:**

This abstract shows how to use Finslerian geometry to compute distances on data manifolds.  The basic idea is to compute expected path lengths under a random metric, rather than deterministic path lengths under the expected Riemannian metric.

Broadly speaking, the abstract is not self-contained and seems to assume the reader is familiar with a specific stochastic approach to Riemannian metric learning.  It also does not do a good job articulating the value/interest of the work to the broader community.

**Questions:**

Abstract:  What does “stochastic smooth mapping” mean?  In general, the second sentence seems to describe a specific approach to Riemannian metric learning, but in reality there are many (!) algorithms that use different approaches.

Abstract:  In general, this abstract doesn’t seem useful for readers who aren’t familiar with the specific body of work the authors are building on.

Page 1, “that links the latent space to the data space” --- shouldn’t f map in the opposite direction usually?  Or, more broadly, what is “f” here?  Please define.  Is f a Riemannian metric or just a map into a latent space?  In the latter case, why is this Riemannian at all?  It seems the intro here needs another paragraph or two to set the stage more clearly for readers who aren’t familiar with the immediate previous work.

It may be useful for convenience to define “Finsler metric” here.

It seems lots of notation is undefined in Proposition 3 (and others).

Figure 2:  Where is “indicatrix” defined?   (Found it later in the Appendix but this abstract does not stand alone.)

Proposition 5: What is omega?

The discussion at the end does a slightly better job than earlier parts of the abstract in setting the stage for this work.

What conclusion should I draw from the experiments in Appendix C?  Is it a good thing or a bad thing that the Finsler and Riemannian geodesics coincide?  Does this affect how I should do Riemannian data analysis?  More discussion is needed here.


**Limitations:**

See above.

**Recommended Decision:**

1: Reject

**Relevance:**

3: Solid fit

**Strengths And Weaknesses:**

The work directly builds on Eklund and Hauberg 2019, and proponents of that work may find additional insight by reading this abstract.  The mathematical proofs here seem reasonable, so I don't think anything in this abstract is incorrect.

The main weaknesses are (1) the exposition is far from self-contained and (2) the abstract (also the accompanying experiments) does not do a reasonable job articulating why the discussion here is relevant to the community.

**Submission Track:**

Extended Abstract (4 Page)

---

> ### Author Response · Authors · 2022-10-27
> **Response to Reviewer ssry**
>
> We would like to thank the reviewer for their feedback that improved the readability and accessibility of the extended abstract. In particular, the abstract has been partially changed, and the propositions are also more detailed, with the terms explicitly defined.
>
> * `Page 1, “that links the latent space to the data space” --- shouldn’t f map in the opposite direction usually? Or, more broadly, what is “f” here? Please define. Is f a Riemannian metric or just a map into a latent space? In the latter case, why is this Riemannian at all? It seems the intro here needs another paragraph or two to set the stage more clearly for readers who aren’t familiar with the immediate previous work.`
>
> $f:\mathbf{R}^q \to \mathbf{R}^D$ is considered to be an immersion. In practice, it can be a GP-LVM, or the decoding part of a VAE, and in both cases, $f$ is stochastic. We pullback the Euclidean metric through this function $f$, and we obtain a (stochastic) Riemannian metric: $g: ({\mathcal T_x \mathcal{M}}, {\mathcal T_x \mathcal{M}}) \to \mathbf{R}_{+}: (u,v) \to u^{\top} J^{\top} J v$, with $J$ the Jacobian of the function $f$. The introduction of the abstract has been modified to make this clearer.
>
> * `What conclusion should I draw from the experiments in Appendix C? Is it a good thing or a bad thing that the Finsler and Riemannian geodesics coincide? Does this affect how I should do Riemannian data analysis? More discussion is needed here.`
>
> This is a good question, and we gave an answer in the Discussion: Yes, it is a good thing that the Finsler and Riemannian geodesics coincide. While we argue it seems more sensible to use the Finsler metric, the Riemannian metric is much more convenient to manipulate and to compute. In conclusion, when working with high-dimensional data, we would recommend to use the Riemannian metric to navigate the latent space of a generative model, since it is a good approximation of the Finsler metric.

---

### Official Review · Reviewer_vF1Q · 2022-10-19
**Riemannian metric == Finsler metric in some LVMs**

**Confidence:** 4
**Soundness:** 3
**Presentation:** 3
**Contribution:** 3
**Overall Rating:** 6

**Summary:**

This extended abstract summarizes work that provides sound theoretical backing for the use of the expected Riemannian metric in the latent space in certain generative models. More specifically, if we assume that the data is mapped to a lower dimensional manifold (by a generative model), we need to be able to compute lengths and other quantities of interest in this latent space. However, since the mapping is stochastic (injectivity is also an issue, which doesn't seem discussed), it does not define a usable Riemannian metric in the latent space. Tosi et al (2014), a paper that is central to this work, considered the expected Riemannian metric instead, and used it in the context of certain latent variable models.

Taking a lead from, but diverging from the work of Tosi el al, the authors focus not on the metric tensor directly, but rather on the geodesics themselves. The main results show that the expected length (since the geodesic distance will also be stochastic) defines a Finsler metric (which does not induce a Riemannian metric). Furthermore, it is shown that with increasing data dimension the Riemannian and Finsler metrics also converge. This equivalence is useful because it provides a justification for the use of the Reimannian metric in such settings, which is also better understood computationally. The results are also shown for the case of Gaussian processes. Other findings, which are both justified theoretically and empirically, in areas of low variance on the data, again both the Finsler and Riemannian metrics are identical.

**Questions:**

A somewhat orthogonal question:
- I have followed this line of work since the paper of Tosi et al, and find it mathematically interesting and also clean and well-motivated. However, one question I have always had is how could some of this be extended to deep neural networks? Are there relevant works that do so? Note that the issue becomes a little complicated in the DNN case since the mappings need not be reversible (and thus the latent space might not be a manifold at all), but that aside.

**Limitations:**

Yes

**Recommended Decision:**

3: Accept

**Relevance:**

3: Solid fit

**Strengths And Weaknesses:**

Strengths:
- Very clearly written and straightforward motivation.
- The theoretical contributions are interesting and valuable: they provide some justification for the use of Riemannian metrics in some LVMs, and also give a better understanding of the quantities of interest in the latent spaces of such models.

"Weaknesses":
- A minor comment: In the extended abstract it is sometimes mentioned "this is a popular method", "commonly used" in reference to some of the work cited. I would beg to differ with these characterizations. I am quite a fan of some of the work referenced, but it is still a niche in many ways. The authors might want to correct for this, and simply mention it in neutral terms. But again, this is just a suggestion.

**Submission Track:**

Extended Abstract (4 Page)

---

> ### Author Response · Authors · 2022-10-27
> **Response to Reviewer vF1Q**
>
> We would like to thank the reviewer for the careful reading of our extended abstract and the constructive feedback.
>
> * `In the extended abstract it is sometimes mentioned "this is a popular method", "commonly used" in reference to some of the work cited. I would beg to differ with these characterizations.`
>
> We also agree with your feedback, and the text has been changed in the camera-ready version to reflect those changes.
>
>
> * `I have followed this line of work since the paper of Tosi et al, and find it mathematically interesting and also clean and well-motivated. However, one question I have always had is how could some of this be extended to deep neural networks? Are there relevant works that do so? Note that the issue becomes a little complicated in the DNN case since the mappings need not be reversible (and thus the latent space might not be a manifold at all), but that aside.`
>
> It is a very good question. Arvanatidis et al [1] have studied the same Riemannian metric in the case of NNs: the decoder part of the VAE, similar to the GP-LVM framework, is considered to be a stochastic immersion. As long as your neural network can be approximated by an immersion, you can always define a Riemannian metric (See Proposition 13.9, in [2]). Note that an immersion does not need to be bijective: it only needs to be a smooth mapping such that its derivatives are injective for every point on the manifold. And, in practice, such a work has been recently used in robotic [3] with the difference that the authors pullback an ambient Riemannian metric, instead of the Euclidean metric.
>
> ---
> [1] Arvanitidis, Georgios, Lars Kai Hansen, and Søren Hauberg. "Latent space oddity: on the curvature of deep generative models." arXiv preprint arXiv:1710.11379 (2017).
>
> [2] Lee, John M. "Smooth manifolds." Introduction to smooth manifolds. Springer, New York, NY, 2013. 1-31.
>
> [3] Beik-Mohammadi, Hadi, et al. "Learning riemannian manifolds for geodesic motion skills." arXiv preprint arXiv:2106.04315 (2021).

---

### Decision · Program_Chairs · 2022-10-21

Accept (Poster)